# The Ultrasensitive Detection of Aflatoxin M_1_ Using Gold Nanoparticles Modified Electrode with Fe^3+^ as a Probe

**DOI:** 10.3390/foods12132521

**Published:** 2023-06-28

**Authors:** Xiaobo Li, Miao Zhang, Haizhen Mo, Hongbo Li, Dan Xu, Liangbin Hu

**Affiliations:** 1Department of Chemistry and Chemical Engineering, Henan Institute of Science and Technology, Xinxiang 453003, China; xbli@hist.edu.cn (X.L.); zmm949798@126.com (M.Z.); 2School of Food Science and Engineering, Shaanxi University of Science and Technology, Xi’an 710021, China; mohz@sust.edu.cn (H.M.); hongbo715@163.com (H.L.); xudan@sust.edu.cn (D.X.)

**Keywords:** aflatoxin M_1_, electrochemical detection, gold nanoparticles, Fe^3+^

## Abstract

The increasing incidence of diseases caused by highly carcinogenic aflatoxin M_1_ (AFM_1_) in food demands a simple, fast, and cost-effective detection technique capable of sensitively monitoring AFM_1_. Recent works predominantly focus on the electrochemical aptamer-based biosensor, which still faces challenges and high costs in experimentally identifying an efficient candidate aptamer. However, the direct electrochemical detection of AFM_1_ has been scarcely reported thus far. In this study, we observed a significant influence on the electrochemical signals of ferric ions at a gold nanoparticle-modified glassy carbon electrode (AuNPs/GCE) by adding varying amounts of AFM_1_. Utilizing ferricyanide as a sensitive indicator of AFM_1_, we have introduced a novel approach for detecting AFM_1_, achieving an unprecedentedly low detection limit of 1.6 × 10^−21^ g/L. Through monitoring the fluorescence quenching of AFM_1_ with Fe^3+^ addition, the interaction between them has been identified at a ratio of 1:936. Transient fluorescence analysis reveals that the fluorescence quenching process is predominantly static. It is interesting that the application of iron chelator diethylenetriaminepentaacetic acid (DTPA) cannot prevent the interaction between AFM_1_ and Fe^3+^. With a particle size distribution analysis, it is suggested that a combination of AFM_1_ and Fe^3+^ occurs and forms a polymer-like aggregate. Nonetheless, the mutual reaction mechanism between AFM_1_ and Fe^3+^ remains unexplained and urgently necessitates unveiling. Finally, the developed sensor is successfully applied for the AFM_1_ test in real samples, fully meeting the detection requirements for milk.

## 1. Introduction

Aflatoxins, produced by *Aspergillus flavus* and *Aspergillus. parasiticus*, which are commonly found in cereal grains, dairy products, beans, and nuts [1,2,3], are toxic compounds with a difuran ring structure. The improper storage conditions can result in contamination with aflatoxins, and the intake of aflatoxins is associated with a significant portion of hepatocellular carcinoma cases worldwide such as impaired liver function and immune response and an increase in susceptibility to infectious diseases [4,5,6]. These toxins can enter the bloodstream and undergo metabolization in the human body. There are four generally recognized aflatoxins designated B_1_, B_2_, G_1,_ and G_2_. Aflatoxin B_1_ (AFB_1_) is the major mycotoxin produced by most species under culture conditions and is the most frequently studied of the four. However, the index compound of AFB1 is not carcinogenic before it is metabolically activated. AFM_1_, a hydroxylated metabolite of AFB_1_ in human food and animal feed, is excreted in urine and secreted in milk in mammalian species within 12 h after consumption, while its toxicity is much less known. AFM_1_, in particular, is reclassified as a naturally occurring carcinogen belonging to Group 1, with the formation of DNA adducts [7]. Nonetheless, many nations have set regulatory limits for maximum allowable AFM_1_ in milk and other dairy products. In 2005, the Food and Drug Administration (FDA) set an AFM_1_ action level in milk and other dairy products at 0.5 μg/L. The European Union (EU) set a much stricter standard that allows a maximum of 0.05 μg/L in milk in 2006 [7]. Turna noted that a high AFM_1_ level in milk was associated with a high level of AFB_1_ in animal feed, which upon consumption could harm both animal and human health [8]. Accordingly, timely monitoring of aflatoxins during biological transformation can help reduce the risk of diseases [9]. Consequently, the development of a sensitive detection method has become a prominent research focus in recent years.

Currently, the identification and quantification of aflatoxins are commonly performed using thin-layer chromatography (TLC) [10], immunoaffinity chromatography [11], high-performance liquid chromatography (HPLC) [12], and enzyme-linked immunosorbent assay (ELISA) [13]. However, these methods have drawbacks such as being time-consuming, requiring special equipment, or involving cumbersome sample pretreatment and false-positive results [14]. A reliable and promising alternative approach that offers high sensitivity, ease of operation, fast analysis, and cost-effectiveness is the electrochemical method. Recently, electrochemical aptamer-based sensors have gained significant attention for aflatoxin monitoring. However, the instability of the biological recognition piece limited the use as anticipated.

Abnous et al. [15] reported an electrochemical sensing strategy for the detection of AFB_1_ based on aptamer-complementary strands of aptamer complex, forming a π-shape structure on the electrode surface, with a detection limit of 2 pg/mL. Ahmadi et al. [16] developed a pencil graphite electrode modified with reduced graphene oxide and gold nanoparticles for the detection of AFM_1_, achieving a detection limit of 0.3 ng/L. Furthermore, the detection of AFM_1_ has been accomplished using voltammetric biosensors, utilizing silver nanoparticles dispersed on an α-cyclodextrin-GQDs nanocomposite [17]. Aflatoxins can be electrochemically oxidized to ketone because of containing an alcohol group, which is generated by the hydrolysis of the aromatic ester group in a basic medium. This allows the direct detection of aflatoxins without the need for recognition elements or tags [18]. Gevaerd et al. [19] reported the direct determination of AFB1 in 2020 at the screen-printed electrode (SPE) modified with gold nanoparticles and graphene quantum dots (AuNPs-GQDs), which exerted an electrocatalytic effect on the oxidation of AFB_1_ (shift of the oxidation peak to less positive values). The performance in ng/mL level by this approach was quite similar to those obtained with other systems where bioreceptors are used. However, the low specificity of this format of detection limits its further application for the selective determination of aflatoxins. Thus, there is still a growing demand for novel sensors that offer a simple electrode modification process, high sensitivity, low cost, and ease of use for the detection of Aflatoxins replacing commonly used biological recognition systems described above.

Aflatoxins belong to the class of coumarin compounds characterized by their difuran ring structures. Previous reports have identified coumarin compounds as fluorescent probes for various metal ions such as Mg^2+^ [20], Cu^2+^ [21,22], Zn^2+^ [23], and others. In recent years, numerous small molecule fluorescent probes for Fe^3+^ have been developed [24,25] based on a selective binding approach for example complexation or chelation. Wang et al. [26] reported a highly selective coumarin-based chemosensor for the detection of Fe^3+^ where coumarin FB displayed a high affinity to Fe^3+^ resulting in forming an FB-Fe^3+^ complex. Zhao et al. [27] specifically designed and synthesized Schiff base probes using phenanthro [9,10-d] imidazole-coumarin derivatives, demonstrating the formation of a 1:1 complex between these probes and Fe^3+^.

Considering this, the electrochemical signals of Fe^3+^ could potentially reflect the concentration of aflatoxins. There appear to be interactions between Fe^3+^ and aflatoxins, which warrant further comprehensive investigation, particularly in understanding the nature of the interaction between Fe^3+^ and aflatoxins.

As is known, electrode modifiers having good conductivity and catalytic activity play an important role in influencing the sensitivity and capability of modified electrodes. Commonly, conducting polymers, molecularly imprinted materials and some metals such as gold, iron, silver, and palladium can be used as electrode modifier materials for enhancing the peak currents, which is necessary for determining the trace amount of analytes in real samples. In this work, we have compared the electrochemical activity of polythionine, molecularly imprinted L-cysteine, and electrodeposited gold nanoparticles (AuNPs). Among them, the uniform deposition of AuNPs onto the glassy carbon electrode (GCE) surface is well known for its ability to increase the effective area and confer the direct electron transfer between the analyte and the electrode base [28,29], making it an excellent sensing platform and giving a much better electrochemical signal towards Fe^3+^ as shown in Appendix A. In this study, we have successfully developed an electrochemical sensor by modifying a GCE with AuNPs (AuNPs/GCE) for the detection of AFM_1_. While the electrochemical synthesis of AuNPs without the requirement of an external linker or functionalizing ligand is well established, aiming for the best electrochemical performance; herein, we have particularly optimized the electrochemical sweeping methods and parameters, deposition times, and minimal solution preparation because the in situ tailoring of nanoparticle surface chemistry resulted in improved catalytic activity and selectivity. The sensor utilizes ferricyanide as a mediator, where the presence of AuNPs greatly facilitates the electrode reaction and enhances the catalytic activity towards ferricyanide. Consequently, this sensor exhibits an unprecedented lowest detectable concentration of AFM_1_ over the widest linear range reported thus far.

To demonstrate the specificity of Fe^3+^ in AFM_1_ detection, an immunoassay column was utilized. Remarkably, an impressive reaction ratio of AFM_1_ to Fe^3+^ was obtained. The performance of the developed sensor was evaluated by measuring AFM_1_ in spiked milk samples, yielding satisfactory analytical results since it is easy, quick, and does not involve developing the biological material methodology.

## 2. Materials and Methods

### 2.1. Chemicals and Apparatus

AFM_1_ was obtained from Toronto Research Chemicals, while HAuCl_4_·3H_2_O (≥99%) was acquired from Sigma Aldrich. KCl, K_3_[Fe(CN)_6_], K_4_[Fe(CN)_6_], NaH_2_PO_4_, and Na_2_HPO_4_ were purchased from Aladdin Reagents. Unless otherwise specified, all reagents were used as received. Phosphate buffer solutions (PBS) were prepared by diluting 0.1 M NaH_2_PO_4_ and 0.1 M Na_2_HPO_4_ stock solutions. All solutions were prepared using double-distilled water with a resistivity of 18 MΩ·cm.

The electrochemical experiments were conducted at room temperature using a CHI900D workstation (Shanghai CH Instrument Ltd., Shanghai, China) equipped with a conventional three-electrode system. The system consisted of a glassy carbon electrode (GCE, 3.0 mm in diameter) as the working electrode, a platinum wire as the counter electrode, and an Ag/AgCl (saturated KCl) electrode as the reference electrode. Electrochemical impedance spectroscopy (EIS) measurements were performed in PBS containing 5 mM K_3_[Fe(CN)_6_]/K_4_[Fe(CN)_6_] (mole ratio of 1:1) and 0.1 M KCl at room temperature. An AUTOLAB PGSTAT302N (Metrohm Auto lab B.V., Herisau, Switzerland) was used for EIS measurements, employing a formal potential of 0.2 V, 5 mV amplitude, and a frequency range from 0.1 Hz to 100 kHz. Nyquist plots were generated from the impedance data and fitted using AUTOLAB Nova 1.8.

Scanning electron microscopy (SEM) experiments were conducted using an XL30 ESEM-FEG (FEI Company, Hillsboro, OR, USA) with an acceleration voltage of 20.0 kV. Molecular fluorescence spectra were measured using an F-180 fluorescence spectroscope (Tianjin Gangdong Co., Ltd., Tianjin, China). The steady-state and transient-state fluorescence spectra were obtained using an FLS 1000 spectrometer (Edinburgh Instruments, West Lothian, UK). Particle size distribution analysis was obtained using a nanoparticle size analyzer Winner 802 (Jinan Weina Particle Instruments, Jinan, China). The enzyme-linked immunosorbent assay (ELISA) was performed using a Varioskan™ LUX (Thermo Fisher Scientific, Waltham, MA, USA).

### 2.2. Preparation of AuNPs/GCE

Figure 1A depicts the schematic diagram illustrating the modification of GCE with AuNPs. To achieve this, the GCE was initially polished to a mirror finish using 0.05 μm alumina slurry on a microcloth. Subsequently, it was ultrasonicated with distilled water for 1 min to ensure cleanliness. The polished and cleaned GCE was then immersed in a solution containing 5 mM HAuCl4 and 0.1 M KCl. AuNPs were formed using cyclic voltammetry (CV) in a potential range from −0.4 V to 1.2 V, with a scan rate of 10 mV/s for 20 cycles. Figure 1B displays the cyclic voltammograms, which exhibit a monotonically increasing trend of the redox waves, confirming the continuous growth of the AuNPs layer.

Following the AuNPs formation, the AuNPs/GCE was thoroughly rinsed with deionized water and transferred to a 0.1 M PBS solution at pH 7.0. The electrode was scanned until a stable voltammogram was obtained. Figure 1C shows the corresponding SEM image of the AuNPs/GCE, revealing a compact layer with uniformly distributed and smaller AuNPs, providing full coverage.

### 2.3. Sample Preparation

AFM_1_ stock solution was prepared by dissolving an appropriate mass of AFM_1_ in pH 7.0 PBS, and working solutions with different concentrations of AFM_1_ were prepared by diluting the stock solution with buffer. The commercial milk was purchased from a local store. In the experimental procedure, 30 mL of the milk sample was transferred to a centrifuge tube. To remove proteins, 20% TCA (trichloroacetic acid) was added, and the mixture was centrifuged for 5 min. Subsequently, centrifugation was performed at a speed of 6000 rpm for 10 min. The resulting supernatant was then filtered through a 0.22 μm filter and passed through an immunoaffinity column. Finally, the filtered sample was subjected to testing. The possibility and reliability of the method being applied in practice were established in regard to evaluating the recovery rate in actual samples.

## 3. Results and Discussion

### 3.1. Electrochemical Characterization of the AuNPs/GCE

The electrochemical behaviors of AuNPs/GCE were investigated by performing CV (cyclic voltammetry) and EIS (electrochemical impedance spectroscopy) measurements in a 0.1 M PBS solution (pH 7.0) containing 5 mM Fe(CN)6^3−/4−^. Figure 2A clearly shows that AuNPs/GCE exhibits remarkable activity and reversibility, as evidenced by the distinct peak-to-peak separation (ΔEp = ca. 78 mV at 100 mV·s^−1^) and enhanced peak current observed in the CVs, in comparison to the bare GCE. Furthermore, the CVs of AuNPs/GCE remained nearly constant despite variations in the number of ultrasonic cleaning cycles, indicating the high stability of the modifier layer.

Additionally, the Nyquist plots presented in Figure 2B, obtained from the EIS measurements, further support the superior conductivity of AuNPs/GCE when compared to the bare GCE. This enhanced conductivity contributes to the exceptional electrochemical catalytic performance exhibited by AuNPs/GCE without the use of any additional biomolecule as an electrode modifier.

### 3.2. Effect of pH and Scan Rates on Fe^3+^ Signals at AuNPs/GCE in the Presence of AFM_1_

The impact of pH variation on the electrochemical response of Fe^3+^ at AuNPs/GCE in the presence of AFM_1_ was further investigated using different buffer solutions prepared and adjusted to a pH range of 4.0 to 8.0. Protons always exert a significant impact on the reaction speed when being involved in the electrochemical reactions of organic compounds. Figure 3A illustrates the gradual increase in peak currents in differential pulse voltammetry (DPV) with an increase in pH within the range of 4–9. The peak currents reach a maximum at pH 7.0 and then decline, leading to the selection of pH 7 for subsequent experiments. This phenomenon is also aligned with the fact that AFM_1_ is generally more stable in neutral pH. Notably, the absence of proton involvement in the reaction is evident as the peak potential does not exhibit a linear shift with pH.

Figure 3B presents the cyclic voltammograms (CVs) obtained at AuNPs/GCE with different scan rates. It is observed that both the cathodic peak current (Ipa) and anodic peak current (Ipc) are directly proportional to the square root of the scan rates within the range of 10–300 mV·s^−1^. The correlated linear equations can be expressed as Ipa (µA) = 5.63 v^1/2^ (mV·s^−1^) + 2.72, and Ipc (µA) = −5.84 v^1/2^ (mV·s^−1^) − 2.23, respectively, with the consistent regression coefficient (r^2^) of 0.99 (inset of Figure 3B), suggesting a diffusion-controlled redox behavior of Fe^3+^ at AuNPs/GCE according to Randles–Sevcik equation instead of a surface reaction-controlled process. Moreover, the fact that peak potential is nearly independent on the scan rate suggests that the redox reaction is electrochemically reversible.

### 3.3. Electrochemical Response of Fe^3+^ at AuNPs/GCE in the Presence of AFM_1_

Differential pulse voltammetry (DPV) is an effective and rapid electroanalytical technique with lower concentration detection limits. The electrochemical responses of Fe^3+^ were further investigated using the DPV technique, with varying concentrations of AFM_1_ added to the electrolyte. Figure 4 illustrates the findings, where it can be observed that the peak currents of Fe^3+^ decrease as the concentrations of AFM_1_ increase within the range of 1.6 × 10^−21^ to 2.5 × 10^−4^ g/L. A linear regression equation for AFM_1_ of I (µA) = −2.34 lg [AFM_1_] (g/L) + 54.261 with a correlation coefficient of 0.99879 was derived from the data. Each current response was measured three times, yielding a relative standard deviation (RSD) of 4.3%. These results clearly demonstrate the successful application of the developed sensor for ultrasensitive detection of AFM_1_, with the lowest observed concentration of 1.6 × 10^−21^ g/L, surpassing previous reports based on the electrochemical method as listed in Table 1. When the electrode was stored in the refrigerator at 4 °C, the current response remained almost unchanged for about 2 weeks by taking advantage of a highly reliable electrode-preparing process. This raises a crucial question regarding the nature of the reaction occurring between Fe^3+^ and AFM_1_, leading to significant suppression of the electrochemical signals of Fe^3+^ in the presence of AFM_1_. We also confirmed the capability of the present method for the monitoring of AFB_1_-NAC and AFB_1_-lysine, which are another two metabolites from AFB1 as shown in Appendix A.

Furthermore, the potential application of this method for detecting other toxins such as zearalenone (ZEA), ochratoxins (OTA), AFB_2_, and AFG_1_ was investigated. The DPVs obtained for these toxins are shown in Figure 5. Similar phenomena were observed for AFB_2_ and AFG_1_, confirming the method’s suitability for selective detection of aflatoxins.

### 3.4. Chronoamperometric Studies

The diffusion coefficient and catalytic rate constant of Fe^3+^ in the presence of AFM_1_ were calculated from chronoamperometry. From the time-current curve, as shown in Figure 6A, it has been deduced that inversely linear dependency exists between the current and the square root of time as shown in Figure 6B. The slope of the linear equation could be obtained by using the Cottrell Equation:I = nFAD^1/2^Cπ^1/2^t^1/2^
where n is the number of transferred electrons, F is the Faraday constant, A is the proportion of the electrode, D is the diffusion coefficient of active substance, C is the initial molar concentration, and t is the running time. From the resulting slope, the D value was obtained to be 6.463 × 10^−8^ cm^2^·s^−1^. Chronoamperometry was also used to measure the catalytic rate constants from the following Equation:*Icat/Id* = c^1/2^π^1/2^ = π^1/2^(kCt) 
where *Icat* and *Id* were the currents of Fe^3+^ at AuNPs/GCE in the presence and absence of AFM_1_ and Fe^3+^, respectively, γ = kCt is the error function, k is the catalytic rate constant, C is the concentration of AFM_1_ and Fe^3+^, and t is the running time (s). From the slope of the *Icat/Id* vs. t^1/2^ plot, as shown in Figure 6C, the k value was obtained to be 4.8 × 10^−2^ cm^3^·mol^−1^·s^−1^.

### 3.5. Effect of Fe^3+^ Concentration on AFM_1_ Fluorescence Intensity

The fluorescence intensity of AFM_1_ in the presence of Fe^3+^ was observed to decrease as the solution pH values increased, as shown in Appendix A. Although, generally, the fluorescence intensity at neutral pH is higher than that in acidic or base environments, there is competition occurring between Fe^3+^ and H^+^ assumed by Patel-Sorrentino et al., for the explanation of pH effect [35]. Additionally, in order to further explore the response properties of AFM_1_ to Fe^3+^, the fluorescence titration experiment in tris buffer solution was performed with the gradual addition of Fe^3+^ to AFM_1_. The concentration of AFM_1_ was maintained at 1 × 10^−5^ M, while the concentration of Fe^3+^ was over the range from 0 to 5.12 × 10^−6^ M. Figure 7A demonstrates that the fluorescence intensity gradually decreases with increasing Fe^3+^ concentration; to put it another way, the addition of Fe^3+^ leads to a remarkable fluorescence quenching of AFM_1_. Once the concentration reaches 8 mM, the fluorescence intensity remains constant at nearly zero and does not change because of quenching saturation. It can be assumed that they may tend to form polymer-like nano-aggregates [36]. The fitting curve is illustrated in Figure 7B, revealing a concentration ratio of AFM_1_ to Fe^3+^ of approximately 1:936. Transient fluorescence analysis (Figure 7C) confirms that the fluorescence quenching process is predominantly static, as AFM_1_, Fe^3+^, and AFM_1_+Fe^3+^ exhibit high coinciding properties. The second-order fitting curve of transient fluorescence spectra by the ordinary least square method is displayed in Appendix A. The non-linear Equation is expressed as follows:Y = A1 × exp(−x/t1) + A2 × exp(−x/t2) + y0 (r^2^ = 0.99)
where A1 = 3155.94 ± 17.67, t1 = 1250.31 ± 14.00, A2 = 156.62 ± 11.16, t2 = 13,712.75 ± 1427.08, y0 = −3.42 ± 2.37, respectively. A calculated fluorescence lifetime of 12.402 µs by formula t = (A1 × t1^2^) + (A2 × t2^2^)/(A1 × t1 + A2 × t2) proves that the interaction between AFM_1_ and Fe^3+^ is ultrafast.

To determine the specific effect of Fe^3+^, various ions including Na^+^, K^+^, Zn^2+^, Cu^2+^, Ni^2+^, Pb^2+^, Cd^2+^, Mn^2+^, Mg^2+^, and Co^2+^ were investigated for their interference on the fluorescence intensity of AFM_1_ under the same conditions. As demonstrated in Figure 7D, none of these ions caused any significant interference even at higher concentrations reflected by negligible responses of AFM_1_. The fluorescence intensity of AFM_1_ was also examined in the presence of vitamin B_12_, a water-soluble vitamin known for its metal ion content, and heme iron. Similar to the effect of Fe^3+^, the fluorescence intensity gradually decreased with increasing concentration, as depicted in Appendix A. These findings indicate a certain interaction between AFM_1_ and Fe^3+^.

To evaluate the intensity of this interaction, diethylenetriaminepentaacetic acid (DTPA) was employed as a competitor against AFM_1_. As it was, DTPA may form a complexation with Fe^3+^ to release AFM_1_ so as to observe an increase in fluorescence intensity. Despite the strong Fe^3+^-binding ability of DTPA, the addition of DTPA (5 mM) to a mixture of AFM_1_ (4 μg/mL) and Fe^3+^ (3.2 mM) unexpectedly led to a further decrease in fluorescence intensity, as depicted in Figure 8A. This suggests that DTPA can surprisingly enhance the interaction between AFM1 and Fe^3+^. The particle size distribution analysis of AFM_1_+Fe^3+^ in the 420–600 nm range (Figure 8B) compared with Fe^3+^ and AFM_1_ in the range of 500–800 nm and 400–1000 nm, respectively, further confirms their combination.

### 3.6. Determination of AFM_1_ in Milk

To assess the effectiveness and feasibility of the proposed method, AFM_1_ levels in milk were measured. Spike and recovery experiments were conducted by measuring DPV responses in real milk samples with known concentrations of AFM_1_ added. The AFM_1_ concentrations in the milk samples were determined through calibration and are presented in Table 2. In all cases, good recoveries were obtained for AFM_1_ varying from 92.0% to 93.9% considering the level of concentration being analyzed, which is comparatively better than those obtained from spectrofluorimetry and ELISA. These findings strongly demonstrate the practical applicability and reliability of the proposed method.

## 4. Conclusions

In this study, we have introduced a novel approach for detecting AFM_1_, utilizing the electrochemical signals of ferricyanide as a sensitive indicator of AFM_1_ concentrations. Notably, we achieved an unprecedentedly low detection limit of 1.6 × 10^−21^ g/L, surpassing previous reports. Furthermore, the recovery rates of 92.4–93.9% obtained from real sample testing underscore the potential of this method as a reliable screening technique for AFM_1_ detection in food. This approach combines the advantages of nanotechnology, supramolecular recognition techniques, and signal amplification, providing a versatile tool for monitoring aflatoxins. Ongoing investigations aim to elucidate the underlying interaction mechanism between AFM_1_ and Fe^3+^. Since most research on the electrochemical detection of aflatoxins is focused on aptamer immunosensors, this work may open new opportunities for Fe^3+^ as a probe for reversely monitoring coumarin-based small molecules. Meanwhile, this work will provide a beneficial reference for sensing of other toxins in food or pharmaceutical assays.

## Figures and Tables

**Figure 1 foods-12-02521-f001:**
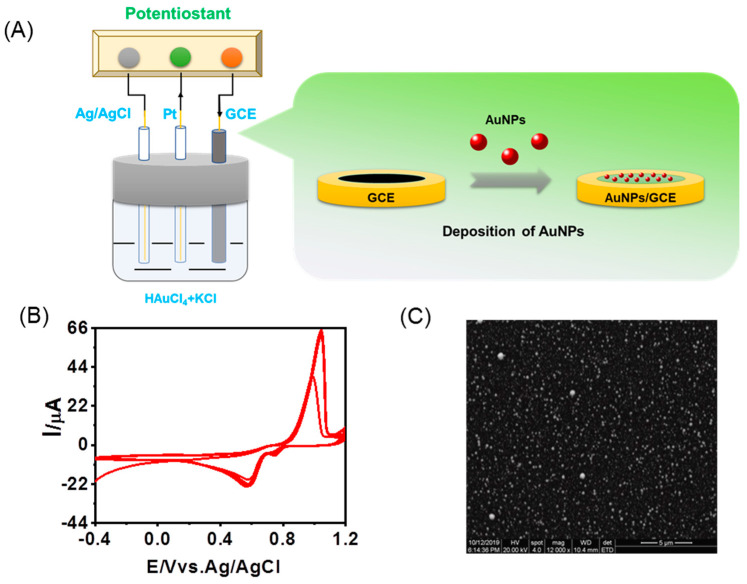
The preparation of AuNPs−modified glassy carbon electrode. (**A**) Schematic diagram of AuNPs modification onto glassy carbon electrode; (**B**) CVs of Au electrodeposition onto GCE in 0.1 M pH 7.0 PBS; (**C**) SEM image of AuNPs/GCE.

**Figure 2 foods-12-02521-f002:**
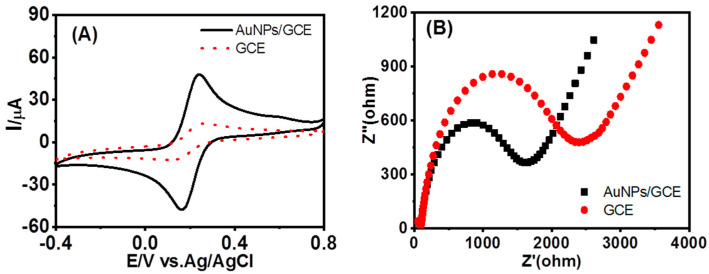
The improved performance of AuNPs−modified glassy carbon electrode. (**A**) CVs of ferricyanide at GCE and AuNPs/GCE in the presence of AFM_1_; (**B**) Nyquist Plot (Z′ vs. −Z′′) of GCE and AuNPs/GCE.

**Figure 3 foods-12-02521-f003:**
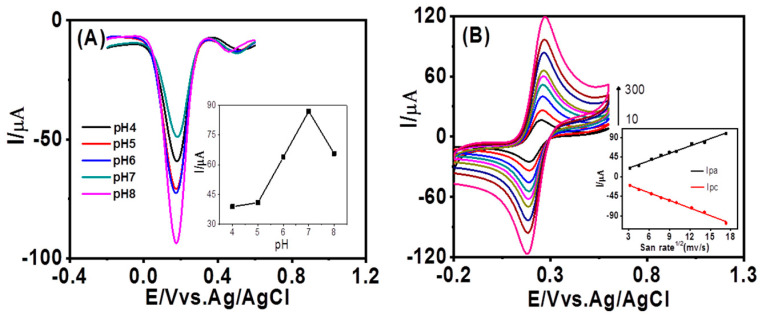
The effects of different pH on the sensitivity of AuNPs/GCE to Fe^3+^ and AFM_1_. (**A**) DPVs of ferricyanide at AuNPs/GCE in 0.1 M PBS in the presence of AFM1 with pH ranging from 4.0 to 8.0. (**B**) CVs of ferricyanide GCE in 0.1 M pH 7.0 PBS at AuNPs/GCE in the presence of AFM_1_ at various scan rates (10, 20, 40, 60, 80, 100, 150, 200, and 300 mV·s^−1^, respectively). Inset shows plots of *Ipa* and *Ipc* (μA) versus the square root of the scan rate.

**Figure 4 foods-12-02521-f004:**
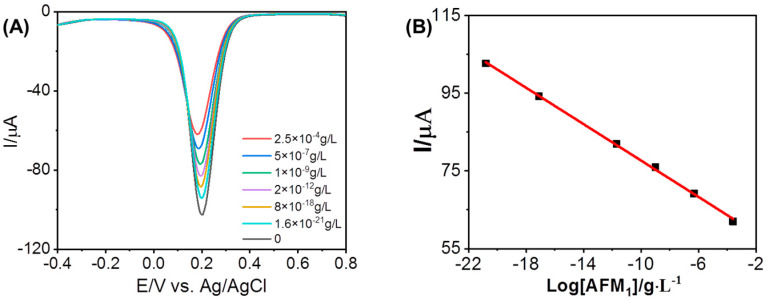
(**A**) DPVs profile of 5 mM ferricyanide at AuNPs/AGCE in the presence of AFM_1_ at different concentrations (2.5 × 10^−4^, 5 × 10^−7^, 1 × 10^−9^, 2 × 10^−12^, 8 × 10^−18^, 1.6 × 10^−21^ g/L, and 0, respectively). (**B**) Inset shows the plot of *Ipa* as a Logarithmic function of the concentration of AFM_1_.

**Figure 5 foods-12-02521-f005:**
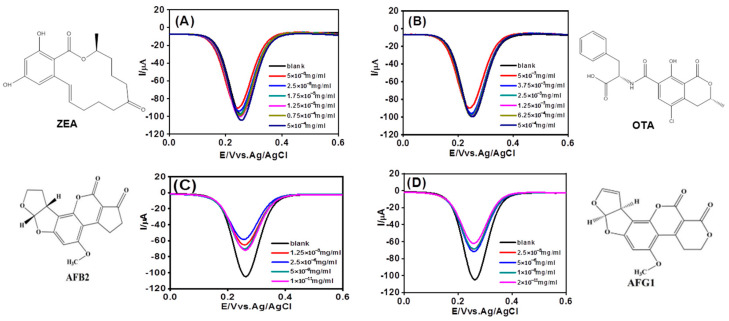
DPVs profile of ferricyanide at AuNPs/GCE in 0.1M PBS in the presence of ZEA (**A**), OTA (**B**), AFB_2_ (**C**), and AFG_1_ (**D**) at different concentrations.

**Figure 6 foods-12-02521-f006:**
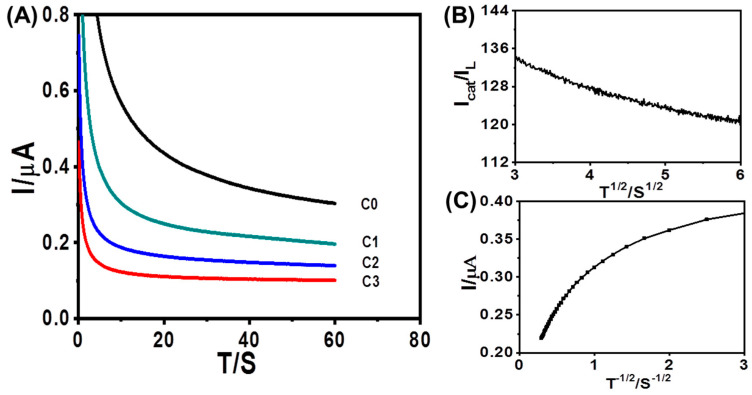
Amperomertric i-t Curves of AFM1 with different concentrations (C0: 0, C1: 6.25 × 10^−10^ g/L, C2:2.5 × 10^−15^ g/L, and C3:1 × 10^−20^ g/L) at AuNPs/GCE in the presence of 5 mM [Fe(CN)6]^4−^. (**A**) Potential is 400 mV. (**B**) Dependence of Icatal/IL on t^1/2^. (**C**) Dependency of transient current on t^−1/2^.

**Figure 7 foods-12-02521-f007:**
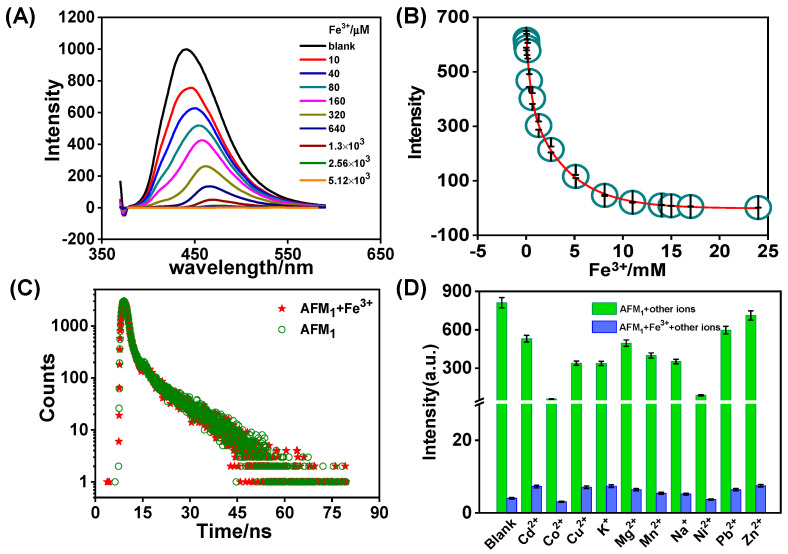
The specific interaction between Fe^3+^ and AFM_1_. (**A**) Effects of Fe^3+^ addition on fluorescence emission of AFM_1_; (**B**) Fluorescence intensity as a function of Fe^3+^ concentration; (**C**) Transient fluorescence lifetime of AFM_1_, AFM_1_ + Fe^3+^; (**D**) Effects of other metal ions on the fluorescence emission of AFM_1_.

**Figure 8 foods-12-02521-f008:**
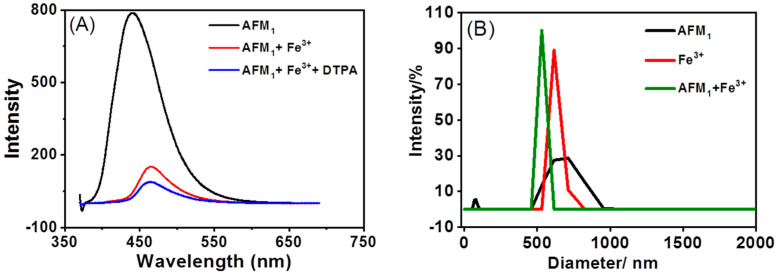
Evaluation on the intensity of interaction between AFM_1_ and Fe^3+^. (**A**) The Fluorescence intensity of AFM_1_, AFM_1_+Fe^3+^, and AFM_1_+Fe^3+^+DTPA. (**B**) The Particle size distribution of AFM_1_, Fe^3+^, and AFM_1_+Fe^3+^.

**Table 1 foods-12-02521-t001:** Comparison of analytical performance for the electrochemical detection of AFM_1_.

System	Detection Limit	Linear Range	Ref.
A-CD-GQDs-AgNPs/GCE	2 μm	0.015–25 μM	[17]
Apt-CS-AuNPs/SPGE	0.9 ng/L	2–600 ng/L	[29]
anti-AFM1/SPGE	2.5 × 10^−8^ g/kg	3 × 10^−8^–1.6 × 10^−7^ g/kg	[30]
Fe_3_O_4_-PANi/IDE	1.98 ng/mL	6–60 ng/mL	[31]
NR/P [5]A-COOH/GCE	0.5 ng/L	5–120 ng/L	[32]
AuNPs/SPE	37 pg/mL	-	[33]
ss-HSDNA-AuNPs/GE	0.36 ng/mL	1–14 ng/mL	[34]
Fe^3+^-AuNPs/GCE	1.6 × 10^−21^ g/L	1.6 × 10^−21^–2.5 × 10^−4^ g/L	This work

**Table 2 foods-12-02521-t002:** The determination of AFM_1_ in milk with three methods.

Method	Add/μM	Detected/μM	Recovery%
This work	2.96	2.78	93.9
	8.91	8.24	92.4
	23.78	22.17	93.2
Spectrofluorimetry	2.2	1.9	86
	3.16	2.5	78
	12.65	10.08	84
ELISA	2.6	2.39	92.1
	8.41	7.82	93
	20.25	18.71	92.4

## Data Availability

The data used to support the findings of this study can be made available by the corresponding author upon request.

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
