# Peer review of "The Ultrasensitive Detection of Aflatoxin M1 Using Gold Nanoparticles Modified Electrode with Fe3+ as a Probe"

_foods, 2023, doi:10.3390/foods12132521_

Round 1
Reviewer 1 Report
The authors reported The ultrasensitive detection of aflatoxin M1 using gold nano- 2 particles modified electrode with Fe2+ as a probe. The experiments have done well and the results are sound. I recommend with minor revision and re review as listed below.
1. The place of procurement of chemicals/equipment must be mentioned.
2. English needs some improvement
3. The figures are not at standards. The x and y scaling should be appropriate. Rescale the numbers with proper intervals.
4. Error bars must be added to the manuscript.
5. The quality of the figures is low.
6. Comparison of this work with other works can be helpful.
7. The voltammetric parameters have to be tested in order to find the best analytical signals of target analytes.
8. The abbreviation for terms should be displayed as it was appeared for the first time.
Needs some improvement
Reviewer 2 Report
1) In case of section 3.2, why the peak current changed with pH? what is the impact of pH?
2) Why the Fe3+ ion peak current decreased in presence of AFM1,? The mechanism should be explained.
The manuscript can be considered for further process after minor revision
Reviewer 3 Report
Journal: Foods
Manuscript ID: 2454377
Title: The ultrasensitive detection of aflatoxin M1 using gold nano-particles modified electrode with Fe2+ as a probe.
Comments
In this work, the authors presented an electrochemical sensing strategy for the detection of a very health concern myotoxin i.e. aflatoxin M1 by using Fe2+ modified Au-NPs. The discovered methods has various significances such as it label free, simple fabrication, and avoided the use of aptamers making is cost effective. Further the electorchemical signal of ferric ion on AuNPs modified glassy carbon electrode has been monitored to measure AMF1. The LoD of the method is found to be in 1.6 pictograms per liter. The method has real time application as revealed by the results in milk sample. The paper has interesting findings and I recommend it for Foods after addressing the following comments.
1. The signal intensity increases with increase in the scan rate as demonstrated by Figure-3 at page lines 159-163. The concentration dependency of signal interferes with the signal intensity enhancement by scan rate. Hence, should there be an optimized scanning rate a higher one? If yes need to mention optimized value.
2. The signal intensity is also decreased by some other interfering toxins such as AFB2 and AFG1 as demonstrated in Figure-5 lines 185. This reveals method has limited selectivity and the interference of other toxins limits the application of method for selective detection of AFM. Justify it in the body of manuscript in section 3.3
3. I recommend identifying the linear range in Figure 7B and mentioning LoD and LoL for fluorescence based assay.
4. It is required to explain the mechanism of FL quenching. And compare this with other mechanism of quenching modes by citing the following relevant references. https://doi.org/10.1016/j.talanta.2019.120316
Language is fine. However, authors can carefully go through it for further improvement.
Reviewer 4 Report
The article is original and very relevant in the field. The authors, developed a sensor which could be successfully applied for the AFM1 detection in milk. The methodology is adequate. I recommend some minor corrections.
Table 1 is split on two pages. You may rearrange the text.
When you write the References, follow the Instructions for authors. Some ref begin with initial of first name, instead of family name. Then, as I know the doi code of the articles is required
The manuscript should be read by a Native English speaker. I ve found minor grammar mistakes.
. Eg- line 56 ease of use- correct as easy to use
